# Caspase-4 promotes metastasis and interferon-γ-induced pyroptosis in lung adenocarcinoma
Yosuke Chiba [1,2], Tomomitsu Doi[2], Kunie Obayashi[2], Kazuhiro Sumida [2], Shohei Nagasaka[2], Ke-Yong Wang[3], Kei Yamasaki[1], Katsuhiro Masago[4], Hirokazu Matsushita[5], Hiroaki Kuroda[6], Kazuhiro Yatera[1] & Motoyoshi Endo [2] ✉

Caspase-4 (CASP4) is a member of the inflammatory caspase subfamily and promotes inflammation. Here, we report that CASP4 in lung adenocarcinoma cells contributes to both tumor progression via angiogenesis and tumor hyperkinesis and tumor cell killing in response to high interferon (IFN)-γ levels. We observe that elevated CASP4 expression in the primary tumor is associated with cancer progression in patients with lung adenocarcinoma. Further, CASP4 knockout attenuates tumor angiogenesis and metastasis in subcutaneous tumor mouse models. CASP4 enhances the expression of genes associated with angiogenesis and cell migration in lung adenocarcinoma cell lines through nuclear factor kappa-light chain-enhancer of activated B cell signaling without stimulation by lipopolysaccharide or tumor necrosis factor. CASP4 is induced by endoplasmic reticulum stress or IFN-γ via signal transducer and activator of transcription 1. Most notably, lung adenocarcinoma cells with high CASP4 expression are more prone to IFN-γ-induced pyroptosis than those with low CASP4 expression. Our findings indicate that the CASP4 level in primary lung adenocarcinoma can predict metastasis and responsiveness to high-dose IFN-γ therapy due to cancer cell pyroptosis.

According to the World Health Organization, primary lung cancer is one of the most frequently occurring cancers globally and the leading cause of cancer-related deaths, accounting for 18.2% of all cancer-related deaths[1]. The 5-year survival rate of patients with non-small cell lung cancer (NSCLC) is only 28.0%[2]. Therefore, the development of more effective lung cancer treatments is a key research imperative.

The tumor microenvironment is known to induce tumor-associated inflammation, e.g., through endoplasmic reticulum (ER) stress signal pathways[3]. Tumor-associated inflammation plays an important role in tumor progression as it promotes both cell growth and metastasis by promoting angiogenesis and increasing tumor cell motility[4,5]. Additionally, tumor-associated inflammation inhibits the antitumor immune response, which is associated with immune checkpoint receptors such as programmed cell death-1 and cytotoxic T lymphocyte-associated protein 4[6]. In recent studies, the response rate in patients with NSCLC treated with immune checkpoint inhibitors was only 19%–34%[7,8]; therefore, there is a need to

develop new treatment strategies. A mechanism underlying the resistance to anti-programmed cell death-1/programmed cell death-L1 antibody therapy is cell death evasion by tumor immune evasion genes, such as *PTPN2*, *ADAR1*, and *TBK1*, which inhibit cell death induced by interferon (IFN)-γ released from activated T cells[9–11]. However, further research is required for an indepth characterization of the mechanisms underlying immunotherapy resistance.

Caspase-4 (CASP4), also known as caspase-11 (Casp11) in mice, is a member of the inflammatory caspase subfamily and promotes inflammation[12,13]. In general, CASP4/Casp11 is activated by inflammatory signals, such as lipopolysaccharide (LPS), and other danger signals to induce inflammatory cell death, which is known as "pyroptosis"[14,15]. ER stress also induces CASP4/Casp11 expression[12]. Additionally, high expression level of CASP4 in cancer cells, including lung cancer cells, is considered a poor prognostic factor[16–19]. In addition, CASP4 is known to promote tumor angiogenesis and cancer cell migration[18,20]. However, the mechanisms

[1]Department of Respiratory Medicine, University of Occupational and Environmental Health, Japan, Kitakyushu, Japan. [2]Department of Molecular Biology, University of Occupational and Environmental Health, Japan, Kitakyushu, Japan. [3]Shared-Use Research Center, University of Occupational and Environmental Health, Japan, Kitakyushu, Japan. [4]Department of Pathology and Molecular Diagnostics, Aichi Cancer Center Hospital, Nagoya, Japan. [5]Division of Translational Oncoimmunology, Aichi Cancer Center Research Institute, Nagoya, Japan. [6]Department of Surgery, Teikyo University Mizonokuchi Hospital, Kawasaki, Japan. ✉e-mail: mendo@med.uoeh-u.ac.jp

underlying the regulation of CASP4 expression, promotion of tumor angiogenesis, and migration of cancer cells in the context of lung cancer remain unknown.

This study demonstrated that CASP4, which is induced by IFN-γ, promotes angiogenesis-related gene expression and tumor cell motility through NF-κB signaling in lung adenocarcinoma cells. Additionally, this is the first study to demonstrate that CASP4 expression increases the susceptibility of lung adenocarcinoma cells to high-dose IFN-γ treatment to induce cell death.

## Results

### CASP4 levels in lung adenocarcinoma correlate with lung cancer progression and prognosis

Initially, we investigated the correlation between CASP4 level and pathological stages using RNA sequencing data of 77 surgical samples of NSCLC collected from the Aichi Cancer Center. CASP4 mRNA expression in lung adenocarcinoma tissues (n = 56) was significantly higher in advanced-stage lung cancer (stages III–IV) than in early-stage cancer (stages I–II) (Fig. 1a and Table. S1), whereas no significant difference in this respect was observed in the 21 non-adenocarcinoma tissues (Fig. S1b). Therefore, we focused our analysis on lung adenocarcinoma patients (56 patients: Fig. S1a and Table S1). Next, we examined the association between CASP4 level and overall survival (OS) in the TCGA dataset. CASP4 level showed no significant correlation with OS in patients with early-stage lung adenocarcinoma (stages I–II) (Fig. S1c). Conversely, patients with advanced-stage lung adenocarcinoma (stages III–IV) with higher CASP4 levels had shorter OS (Fig. S1d). This suggested that CASP4 level might be a poor prognostic factor in patients with advanced-stage lung adenocarcinoma.

### CASP4 in lung adenocarcinoma cells regulates the expression of genes associated with both angiogenesis and cell migration

We established CASP4 overexpressing NCI-H292 and A549 cells to elucidate the role of CASP4 in lung adenocarcinoma cells (Fig. 1b, S1f, and S1g). Both cell lines showed no difference in proliferation compared with mock cells in vitro (Fig. S1e). We compared the RNA sequencing data of CASP4 overexpressing and mock-infected NCI-H292 cells (Fig. 1c). Gene enrichment analysis revealed enrichment of GO terms "angiogenesis" and "positive regulation of cell migration" in genes upregulated in CASP4 overexpressing cells compared with those in mock cells (Fig. 1d, e). Additionally, the expression of antiapoptotic genes, such as BCL2A1 and BCL2L1, was upregulated in CASP4 overexpressing cells compared with that in mock cells (Fig. 1f). These results indicate that CASP4 promotes angiogenesis, cell motility, and survival in lung adenocarcinoma cells. Next, we established CASP4 KO NCI-H292 and CASP4 KD H1975 cells to confirm whether CASP4 KO or KD in lung adenocarcinoma cells reduce the expression of genes associated with angiogenesis and cell migration (Fig. 2a–c and S2a and S2b). Both cell lines showed no difference in proliferation compared with mock cells in vitro (Fig. S2d). We performed RNA sequencing analysis using CASP4 KO cells and mock cells and compared the results, as described previously (Fig. 2d). The expression of genes associated with both "angiogenesis" and "positive regulation of cell migration" was downregulated in CASP4 KO cells compared with that in mock cells (Fig. 2e, f). Among the genes related to angiogenesis and cell migration, seven overlapping genes that were upregulated in CASP4 overexpressing cells and downregulated in CASP4 KO cells were selected (Fig. 2g). Among the differentially expressed genes (DEGs), PTGS2 and EPHA2 were reproducible in RT-qPCR (Figs. 2h, i, S1h, S2c). These results indicate that CASP4 in lung adenocarcinoma cells regulates the expression of genes associated with both angiogenesis and cell migration.

### CASP4 promotes the migration of lung adenocarcinoma cells

Wound healing assay and transwell cell migration assay were performed using CASP4- overexpressing, -KO, and -KD cells to confirm the role of CASP4 in lung adenocarcinoma cell migration, as described previously[18]. In the wound healing assay, compared with mock cells, the migration of CASP4 overexpressing cells resulted in a significantly increased percentage of wound closure area (Fig. 3a, b). On the contrary, the percentage of wound closure area in CASP4-KO or -KD cells was significantly reduced compared with that in mock cells (Figs. 3c, d, S3a, and S3b). The transwell cell migration assay demonstrated significantly reduced migration ability of CASP4-KO or -KD cells (Figs. 3e, f, S3c, and S3d). Furthermore, we rescued PTGS2 and EPHA2, whose expressions were decreased by CASP4 KO or KD, in NCI-H292 mock or CASP4 KO cells. Wound healing assay showed that rescue of each of these two genes restored cell motility in CASP4 KO cells (Figs. 3g–j, and S3e, S3f). These results indicate that CASP4 promotes the migration of lung adenocarcinoma cells.

### Casp11 KO reduces tumor progression and metastasis in the syngeneic and allogeneic mouse model

Mice Casp11 is an ortholog of human CASP4[21]. We established Casp11 KO 3LL cells to investigate the role of Casp11 (Fig. 4a, b). The expression of mouse genes associated with angiogenesis, such as mPtgs2 and mEphA2, was reduced in Casp11 KO 3LL cells compared with that in mock cells, which was also observed in CASP4 KO lung adenocarcinoma cells (Fig. 4c). Next, we established the syngeneic mouse model by subcutaneously transplanting Casp11 KO 3LL cells or mock cells into C57BL/6 N mice (Fig. 4d). The model transplanted with Casp11 KO 3LL cells showed significantly reduced diameters and weights of primary tumors compared with those in the mock model, despite equivalent in vitro proliferation observed between Casp11 KO and mock cells (Fig. 4e–g). Furthermore, lung metastasis was barely detected in the model transplanted with Casp11 KO 3LL cells (Fig. 4h). The expression of CD31 (PECAM-1), a marker of vascular endothelial cells, and CD34, a marker of progenitor cells, in tumors was markedly attenuated in the model transplanted with Casp11 KO 3LL cells compared with that in the mock model (Fig. 4i, j). In addition, to evaluate the potential contribution of Casp11 in shaping the tumor immune microenvironment, we also conducted subcutaneous tumor transplantation in immunodeficient mouse model (BALB/c nu/nu). In immunodeficient mice, Casp11 KO cells also showed decreased proliferation rate, suppression of lung metastasis, and suppression of intratumoral angiogenesis (Fig. S4). These results indicate that Casp11 in mouse lung tumor cells contributes to tumor growth and angiogenesis to promote metastatic potential in mice, regardless of host tumor immunity.

### NF-κB contributes to cell migration- and angiogenesis-associated genes upregulated by CASP4

We used TRRUST[22] to determine the mechanism by which CASP4 regulates cell migration and angiogenesis. The results showed that the genes regulated by NF-κB subunits—NFKB1 and RELA—were significantly upregulated in CASP4 overexpressing cells and downregulated in CASP4 KO cells (Fig. 5a, b). CASP4-KO or -KD cells showed reduced NF-κB activity compared with mock cells (Figs. 5c and S5a). Additionally, compared with mock cells, the phosphorylation level of serine 536 (Ser536) in p65, which is important for NF-κB activation[23], was decreased in CASP4-KO or -KD lung cancer cells (Figs. 5d and S5b). Furthermore, the NF-κB activity and p65 phosphorylation levels were restored after rescue of CASP4 expression in CASP4 KO cells (Fig. 5e, f). Compared with mock cells, nuclear p65 translocation in response to TNF-α was enhanced in CASP4 overexpressing A549 cells; conversely, it was reduced in CASP4 KD cells (Fig. S5c, d). BAY11-7082, which is an NF-κB inhibitor[24], suppressed the induction of PTGS2, SERPINE1, and EPHA2 mRNA by CASP4 (Figs. 5g and S5e). Simultaneously, BAY11-7082 also reduced CASP4 mRNA expression (Fig. 5g). However, it did not reduce CASP4 protein expression (Fig. 5h), suggesting that the suppression of these gene expressions was mainly due to the NF-κB inhibitory effect. Furthermore, BAY11-7082 abrogated the increased cell migration ability of CASP4 overexpressing cells (Fig. 5i, j). These results indicate that NF-κB contributes to cell migration and angiogenesis.

**Fig. 1 | *CASP4* overexpression upregulates angiogenesis, cell migration, and expression of antiapoptosis-related genes. a** Comparison of *CASP4* mRNA expression between early-stage and advanced-stage lung adenocarcinoma tissues (stages I–II vs. III–IV) based on RNA sequencing (RNA-seq) data (n = 56). **b** Representative images of immunoblotting analysis of CASP4 protein in mock and *CASP4*-overexpressing (C4OE) NCI-H292 cells. Bar graphs indicate the band intensities of CASP4 normalized to beta-actin bands (n = 3). Data from mock was set at 1. **c** Flow chart of the RNA-seq experiment and data analysis. **d** The top eight upregulated gene ontology (GO) terms in NCI-H292 cells following *CASP4* overexpression based on GO analysis of coding sequence expression data obtained from RNA sequencing. The horizontal axis shows accumulative hypergeometric *p* values. **e** Representative heatmap of cell migration- and angiogenesis-related genes upregulated by *CASP4*. **f** RNA-seq analysis of antiapoptosis-related mRNAs, such as *BCL2*, *BCL2A1*, *BCL2L1*, and *MCL1*, between C4OE and mock cells. Each gene is presented as fragments per kilobase of exon million mapped reads (FPKM). N.S: nonspecific; n.s.: not significant. Data are expressed as mean ± standard deviation (n = 3). *P < 0.05, **P < 0.01 compared with mock. Densitometry values are shown above each blot.

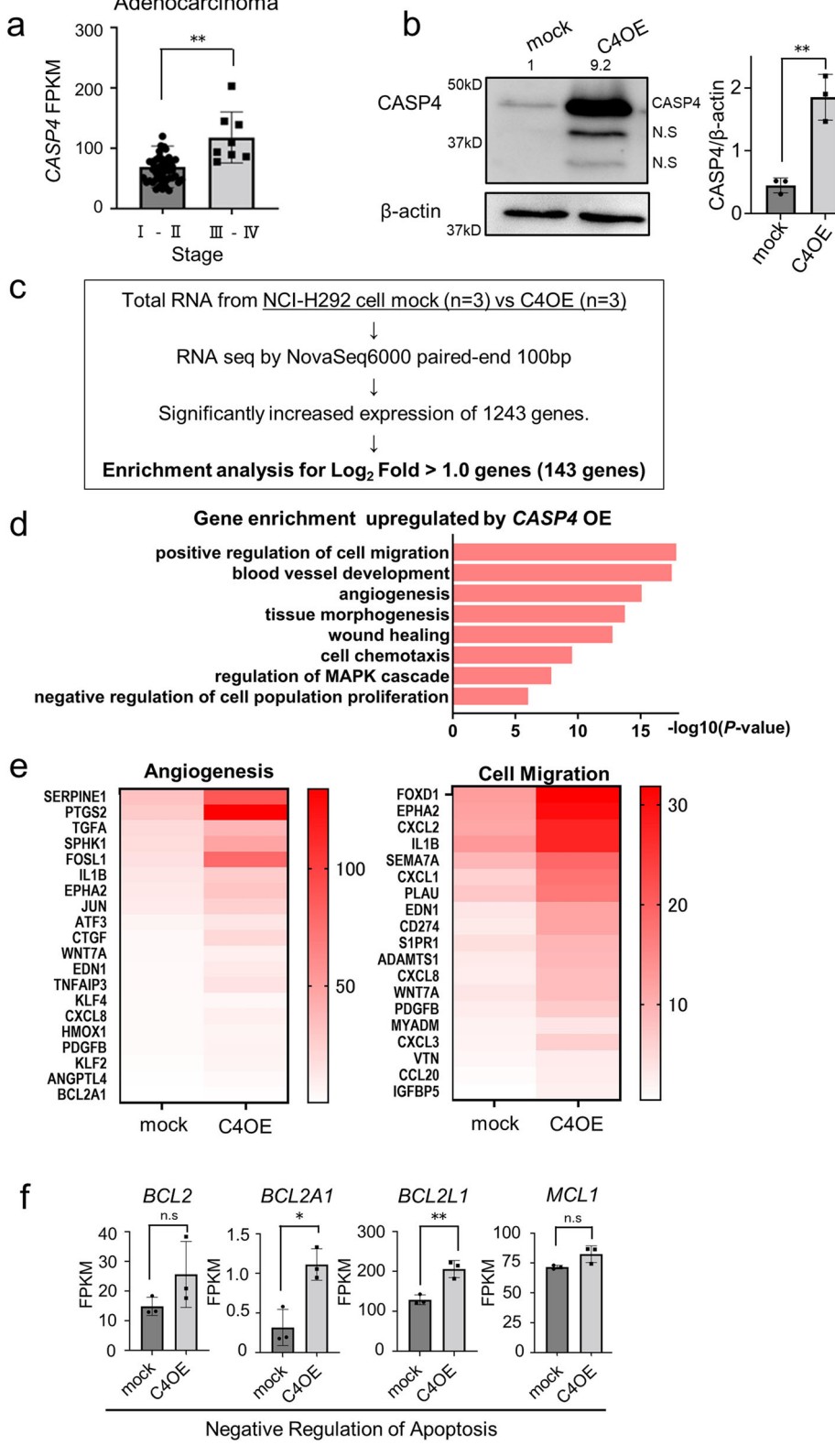

## CASP4 is induced by IFN-γ or ER stress in lung adenocarcinoma cells

We investigated the mechanism of induction of CASP4 in lung adenocarcinoma cells. The top 133 genes coexpressed with *CASP4* in lung adenocarcinoma were selected based on the TCGA dataset (Spearman's correlation coefficient >0.4). Then, using TRRUST, we explored the transcription factors that regulate these genes, including *CASP4* (Fig. 6a). STAT1 was identified as the transcription factor that showed the strongest association with CASP4 (Fig. 6b). Therefore, we stimulated lung adenocarcinoma cells with IFN-γ, which is a major STAT1 activator, and found that *CASP4* was induced by IFN-γ even at low doses that have no cell killing effect (Fig. 6c, d). Next, we investigated the effects of IFN-γ on *CASP4*

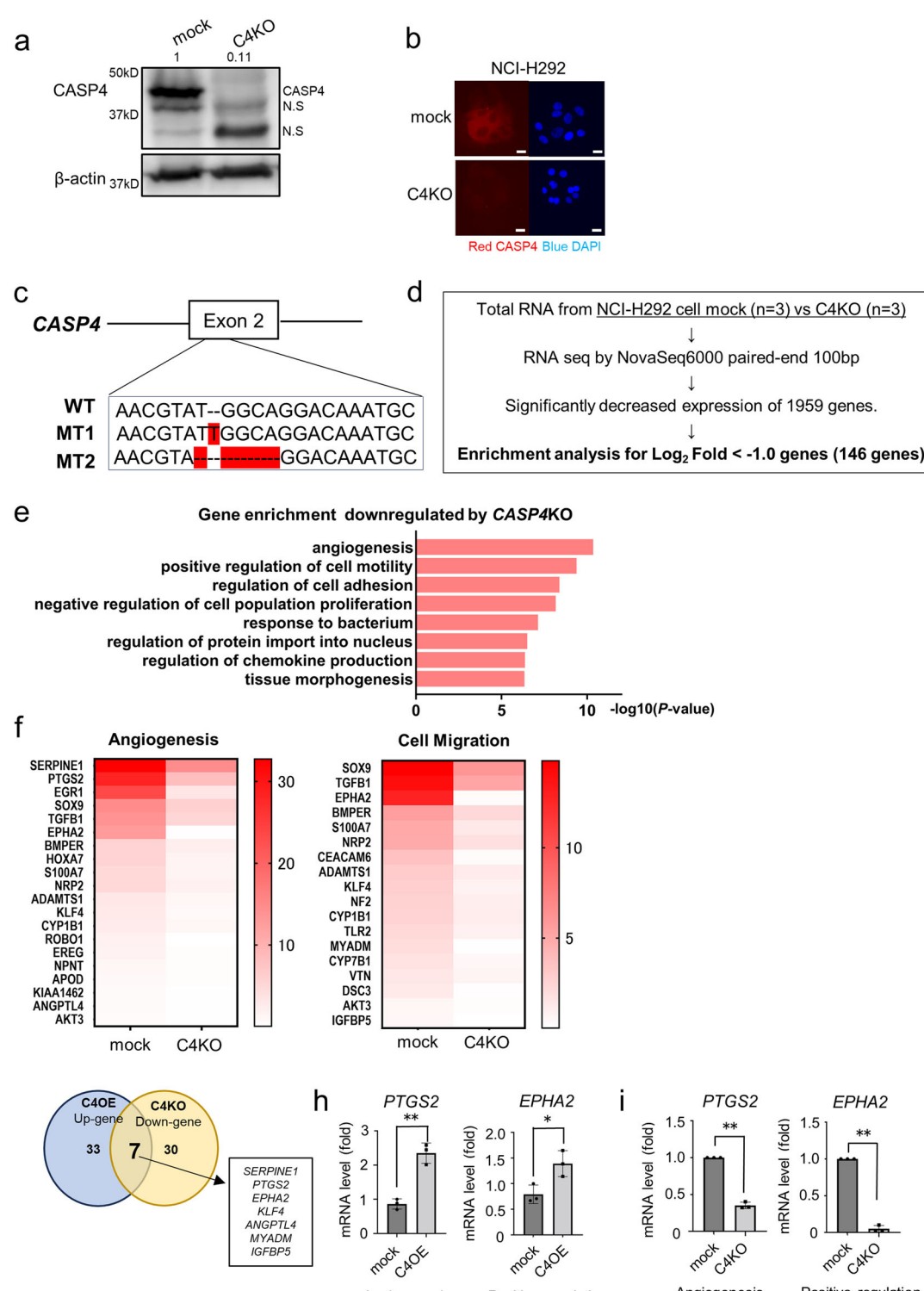

**Fig. 2 | *CASP4* knockout decreases angiogenesis and expression of cell migration-related genes. a** Representative images of immunoblotting analysis of CASP4 protein in mock and *CASP4*-knockout (C4KO) NCI-H292 cells. **b** Representative images of fluorescent immunostaining of CASP4 protein in mock and *CASP4*-knockout (C4KO) cells. Scale bar, 20 μm. **c** Sequence showing insertion and deletion in exon 2 of human CASP4 in C4KO NCI-H292 cells. **d** Flowchart of the RNA-seq experiment and data analysis. **e** The top eight downregulated GO terms in NCI-H292 cells following *CASP4* KO based on GO analysis of CDS expression data obtained from RNA sequencing. The horizontal axis shows accumulative hypergeometric *p* values. **f** Representative heatmap of cell migration- and angiogenesis-related genes downregulated by *CASP4 KO*. **g** Differentially expressed genes that overlapped between C4OE upregulated and C4KO downregulated in "angiogenesis" and "positive cell migration. **h** Positive regulation of cell migration- and angiogenesis-related mRNAs, such as *EPHA2* and *PTGS2*, based on RT-qPCR in C4OE and mock cells. **i** Positive regulation of cell migration- and angiogenesis-related mRNAs, such as *EPHA2*, and *PTGS2*, based on RT-qPCR in C4KO and mock cells." N.S: nonspecific. Data are expressed as mean ± standard deviation (n = 3). **P < 0.01 compared with mock. Densitometry values are shown above each blot.

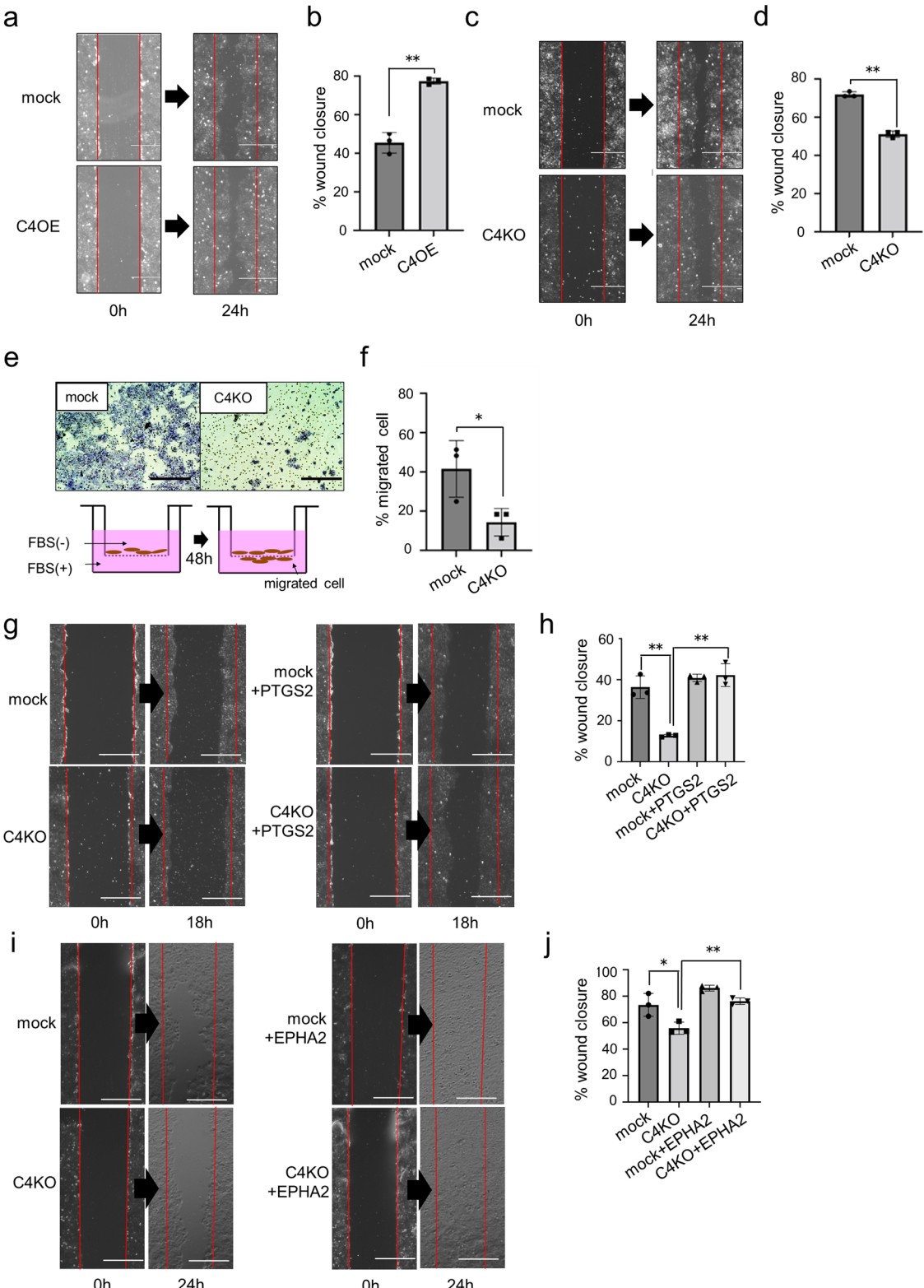

**Fig. 3 | CASP4 promotes cell migration ability in vitro. a** Representative images of the wound healing assay using mock or C4OE NCI-H292 cells at 0 and 24 h (scale bar, 500 μm). **b** The ratios of wound width after 24 h to initial wound width in (**a**) were graphed as percent wound closure (n = 3). **c** Representative images of the wound healing assay using mock or C4KO NCI-H292 cells at 0 and 24 h (scale bar, 500 μm). **d** The ratios of wound width after 24 h to initial wound width in (**b**) were graphed as percent wound closure (n = 3). **e** Representative images of cell migration assay using mock or C4KO NCI-H292 cells at 48 h (scale bar, 100 μm). **f** Quantitative analysis of cell migration in (**e**) by mock or C4KO NCI-H292 cells as indicated by percentage of migrated cells at 48 h (n = 3). **g** Representative images of the wound healing assay using mock and C4KO cells with or without *PTGS2* gene rescue at 0 and 18 h (scale bar, 500 μm). **h** The ratios of wound width after 18 h to initial wound width in (**g**) were graphed as percent wound closure (n = 3). **i** Representative images of the wound healing assay using mock and C4KO cells with or without *EPHA2* gene rescue at 0 and 24 h (scale bar, 500 μm). **j** The ratios of wound width after 24 h to initial wound width in (**i**) were graphed as percent wound closure (n = 3). Data are expressed as mean ± standard deviation (n = 3). *$P < 0.05$, **$P < 0.01$ compared with mock.

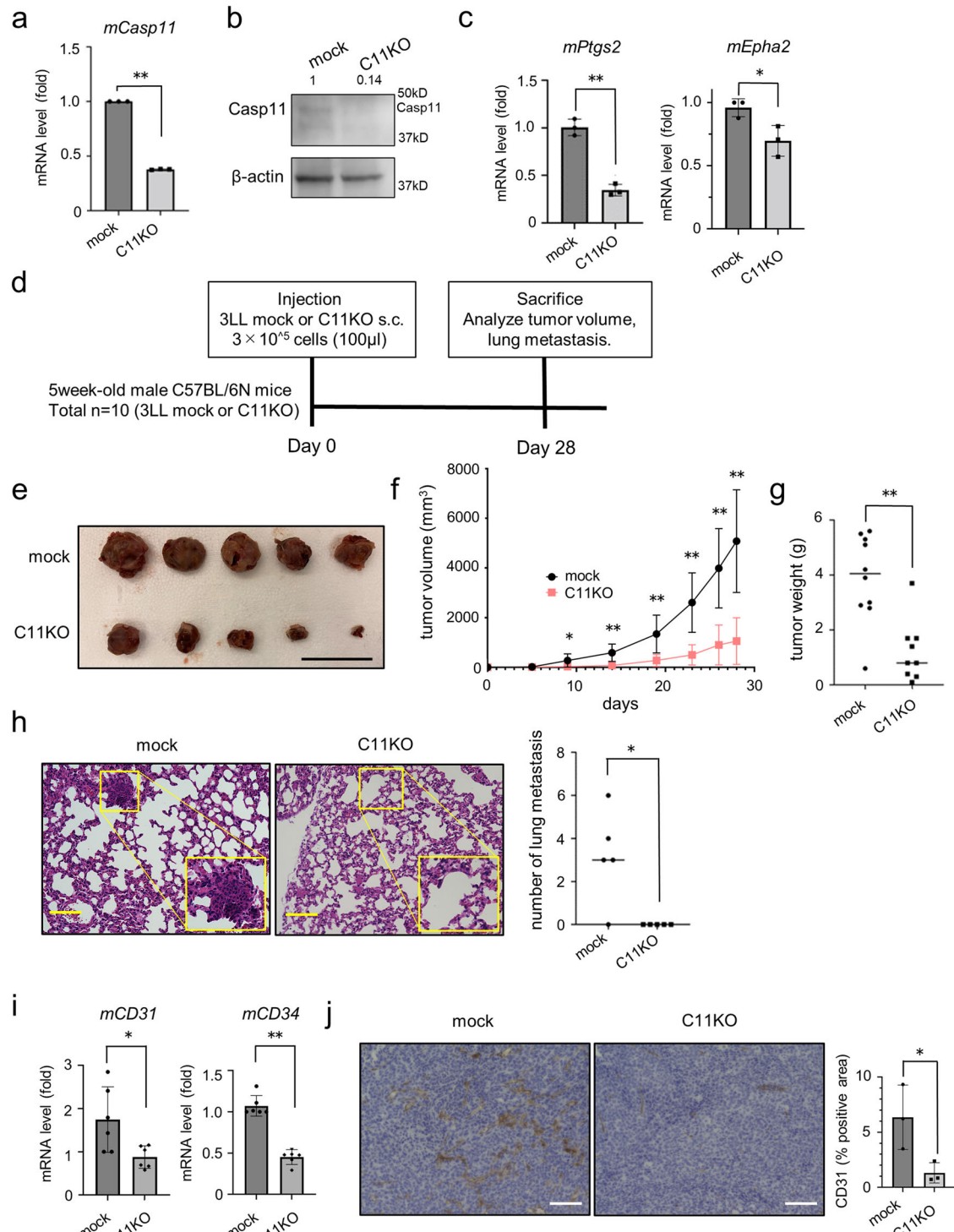

**Fig. 4 | *Casp11* knockout attenuates primary tumor growth and angiogenesis to inhibit metastasis. a** Caspase-11 *(Casp11)* mRNA levels in primary tumors of mock and *Casp11*-KO (C11KO) Lewis Lung Carcinoma (3LL) cells. **b** Representative images of immunoblotting analysis of Casp11 protein in mock and C11KO 3LL cells. **c** mRNA levels of mouse *Ptgs2* and *Epha2* in mock and C11KO 3LL cells. Data from the mock group were set at 1. **d** Diagram showing the time course of the syngeneic mouse model. **e** Representative macroscopic images of primary tumors of mock or C11KO 3LL cells at 28 days after injection (scale bar, 30 mm). **f** Primary tumor volume of mock and C11KO 3LL cells injected in the mouse (n = 10). **g** Primary tumor weight of mock and C11KO cells at 28 days after injection (n = 10).

**h** Representative microscopy images of HE staining of the primary tumors of mock and C11KO cells (left panel). Higher magnifications of the left squared area are indicated on the right image (scale bar, 100 μm). The number of tumor metastases in the left lung in mice injected with mock and C11KO cells (right panel). **i** mRNA levels of mouse *CD31* and *CD34* in mock and C11KO 3LL cells. Data from the mock were set at 1. **j** Representative microscopy images of CD31 immunostaining of the primary tumors of mock and C11KO cells (left panel) (scale bar, 100 μm). Quantitative analysis of CD31 positive area of primary tumors of mock and C11KO cells (right panel). Data are expressed as mean ± standard deviation (n = 3). *P < 0.05, **P < 0.01 compared with mock. Densitometry values are shown above each blot.

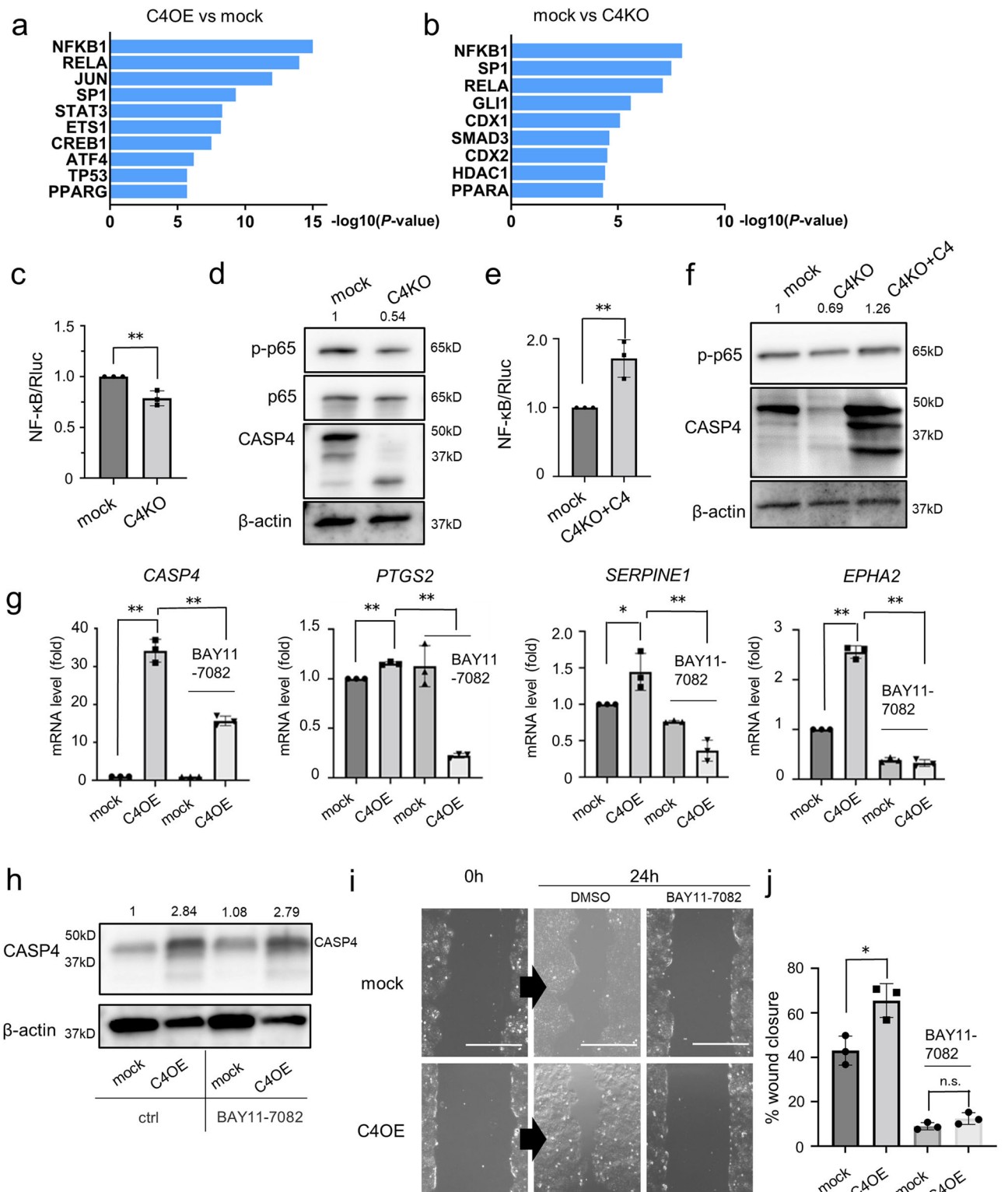

**Fig. 5 | NF-κB pathway contributes to the upregulation of genes related to angiogenesis and cell migration.** Ranking of the top ten upregulated transcription factors presumed to be involved in C4OE (**a**) and C4KO (**b**) by TRRUST database analysis of the CDS expression data obtained from RNA sequencing of NCI-H292 cells. The horizontal axis shows accumulative hypergeometric *p* values. **c** Comparison of relative luciferase activity between mock and C4KO NCI-H292 cells. **d** Immunoblotting analysis of phospho-p65, p65, and CASP4 in mock and C4KO cells. **e** Comparison of relative luciferase activity between mock and C4KO cells with rescued *CASP4* gene (C4KO + C4) (n = 3). **f** Immunoblotting analysis of phospho-p65 and CASP4 protein in mock, C4KO, and C4KO + C4 cells. **g** mRNA levels of *CASP4*, *PTGS2*, *SERPINE1*, and *EPHA2* in mock and C4OE cells with or without treatment with the NF-κB inhibitor BAY11-7082. **h** Protein levels of CASP4 in mock and C4OE cells with or without treatment with BAY11-7082.

**i** Representative images of the wound healing assay with or without treatment with BAY11-7082 using mock or C4OE NCI-H292 cells at 0 and 24 h (scale bar, 500 μm). **j** The ratios of wound width after 24 h to initial wound width in (**i**) were graphed as percent wound closure (n = 3). In each experiment, treatment with BAY11-7082 (2.5 μM) lasted 8 h. n.s.: not significant. Data are expressed as mean ± standard deviation (n = 3). *P < 0.05, **P < 0.01 compared with mock. Densitometry values are shown above each blot.

**Fig. 6 | CASP4 is induced by IFN-γ. a** Identification of the top 133 genes (Spearman's correlation >0.4) coexpressed with *CASP4* using the TGCA-LUAD dataset. **b** The top 133 genes coexpressed with *CASP4* were analyzed via TRRUST analysis to determine the transcription factors involved. The horizontal axis shows accumulative hypergeometric p-values. **c** *CASP4* mRNA induction by IFN-γ stimulation in NCI-H292 cells. Cells were treated for 24 h. **d** Immunoblotting analysis of CASP4 in NCI-H292 cells treated with PBS and 1 ng/mL of IFN-γ. Cells were treated for 48 h. **e** Evaluation of *CASP4* mRNA expression in NCI-H292 cells by RT-qPCR and assessment of cell viability by CCK-8 assay after treatment with IFN-γ at different concentrations. The samples for RT-qPCR and CCK-8 assay were collected after 24 h and 72 h, respectively. **f** Comparison of *CASP4* mRNA expression by RT-qPCR between mock and CASP4 OE after treatment with high-dose IFN-γ (100 ng/mL) for 24 h. **g** Downregulation of *CASP4* mRNA expression by *STAT1* knockdown (shSTAT1) in NCI-H292 cells. **h** Representative images of immunoblotting analysis of CASP4 and STAT1 protein in mock and shSTAT1 NCI-H292 cells. n.s.: not significant; N.S: non-specific. Data are expressed as mean ± standard deviation (n = 3). *$P < 0.05$, **$P < 0.01$ compared with PBS treatment. Densitometry values are shown above each blot.

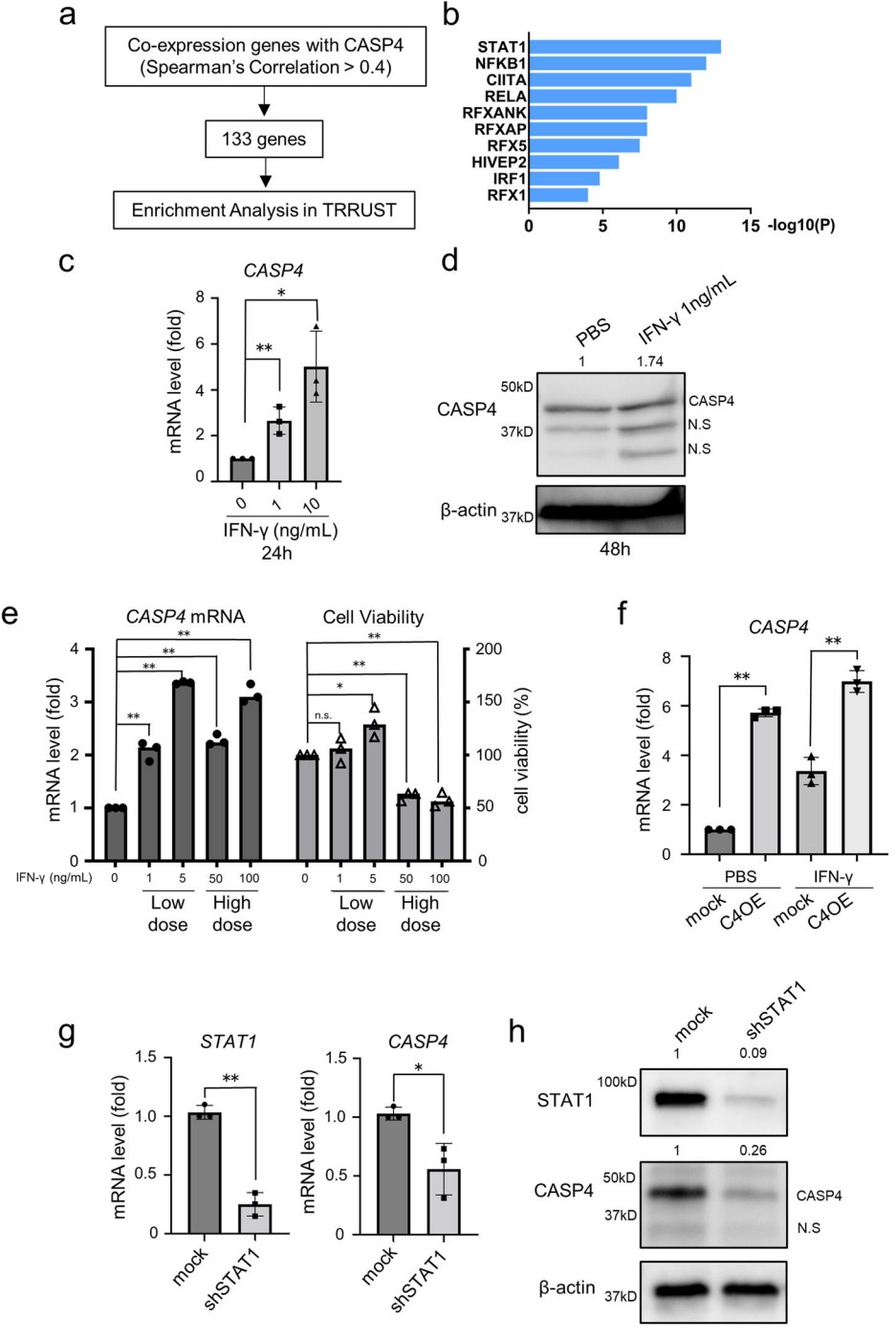

induction and cancer cell viability. Low-dose IFN-γ (1 or 5 ng/mL) slightly increased the cell viability; however, high-dose IFN-γ (50 or 100 ng/mL) significantly decreased the cell viability, suggesting that low-dose IFN-γ enhances *CASP4* expression in lung adenocarcinoma cells without inducing cell death (Fig. 6e). In addition, even when treated with high-dose IFN-γ, *CASP4* overexpressing cells exhibited higher *CASP4* mRNA level compared to mock cells (Fig. 6f). Furthermore, genetic or pharmacological inhibition of STAT1 downregulated the expression of *CASP4* at mRNA and protein levels (Figs. 6g, h and S6c). Moreover, the ER stress inducer tunicamycin induced *CASP4* expression in lung adenocarcinoma cells with the phosphorylation of tyrosine 701 in STAT1, as described previously (Fig. S6a and S6b)[25].

## CASP4 increases the susceptibility of lung adenocarcinoma cells to high-dose IFN-γ treatment-induced pyroptosis

The results indicated that CASP4 induced by IFN-γ is involved in the malignant transformation of lung adenocarcinoma cells. In contrast, IFN-γ is an important factor in the antitumor immune response as it induces tumor cell cycle arrest or cell death[26]. Therefore, we examined the effect of CASP4 expression in lung adenocarcinoma cells on their sensitivity to IFN-γ-induced cell death. The cytotoxic effects of high-dose IFN-γ treatment were found to be more pronounced in *CASP4* overexpressing cells than in mock cells (Fig. 7a–c). Additionally, this enhanced cytotoxic effect was dependent on the enzymatic activity of CASP4 (Fig. 7b). Fluorescence-activated cell sorting and fluorescence microscopic analysis revealed higher propidium iodide-positivity in *CASP4* overexpressing cells than in mock

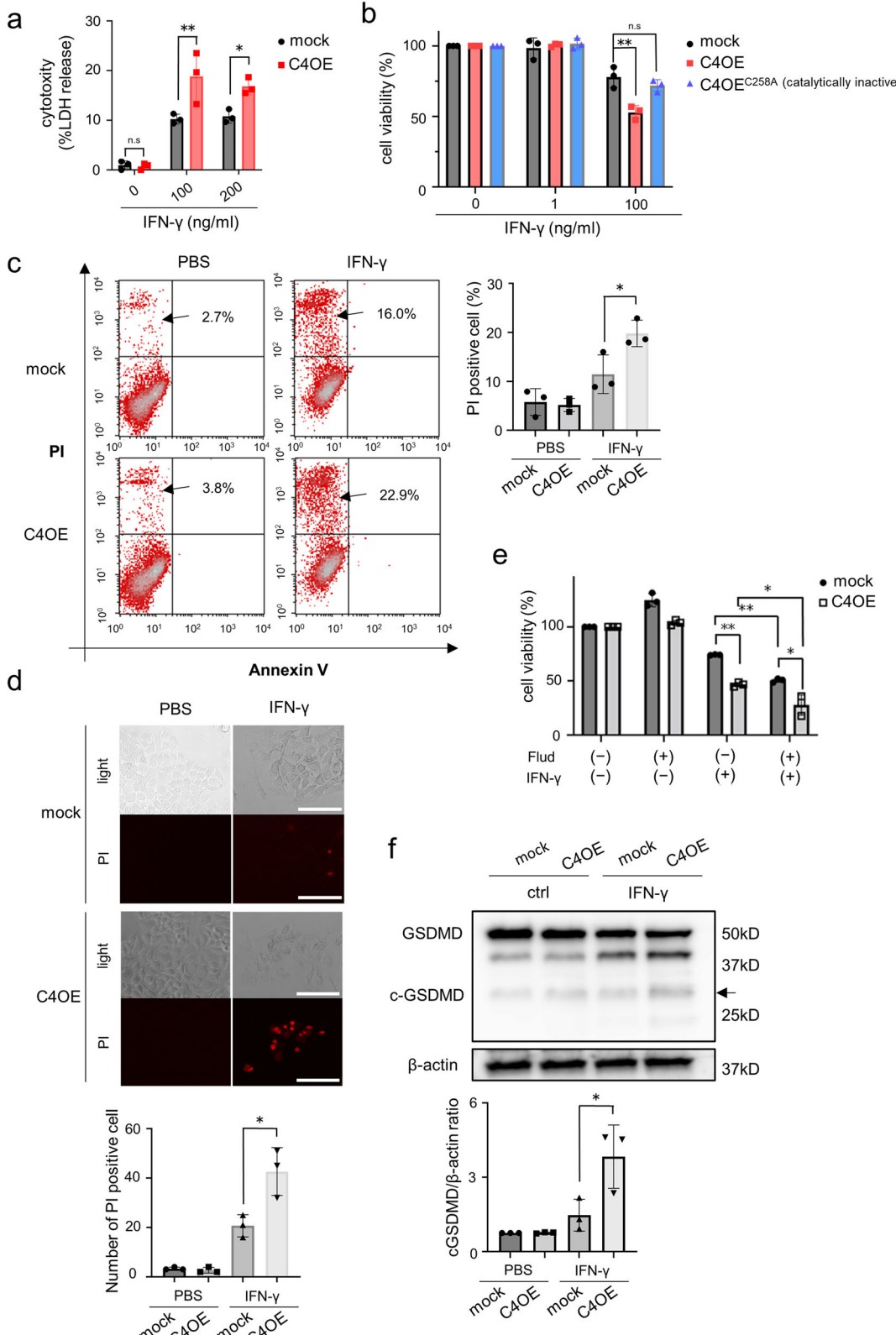

**Fig. 7 | CASP4 expression increases the susceptibility of lung adenocarcinoma cells to IFN-γ treatment to induce cell death depending on CASP4 enzyme activity. a** LDH release from NCI-H292 cells at 5 days after IFN-γ treatment. **b** Viability of mock, C4OE, and C4$^{C258A}$OE NCI-H292 cells treated with 1 or 100 ng/mL IFN-γ. **c** Annexin-V/PI positivity in mock and C4OE cells at 72 h after treatment with PBS or 100 ng/mL IFN-γ, as determined via flow cytometry (left panel). Bar graphs indicate the PI-positive cells (n = 3) (right panel). **d** Representative immunofluorescent images of PI staining of mock and C4OE cells at 72 h after treatment with PBS or 100 ng/mL IFN-γ. Scale bar, 100 μm. (upper panel). Bar graphs indicate the PI-positive cells in high-power field (n = 3) (lower panel). **e** Cell viability of NCI-H292 cells treated with IFN-γ 100 ng/mL for 72 h with or without STAT1 inhibitor, fludarabine (Flud) 5 μM. Cells were pretreated with Flud for 3 h before IFN-γ treatment. **f** Immunoblotting analysis of GSDMD in NCI-H292 cells treated with 100 ng/mL of IFN-γ. Cells were treated for 72 h. n.s.: not significant. Data are expressed as mean ± standard deviation (n = 3). *P < 0.05, **P < 0.01 compared with mock. Densitometry values are shown above each blot.

cells (Fig. 7c, d). Surprisingly, treatment with a STAT1 inhibitor (fludarabine) did not suppress IFN-γ-induced cell death in *CASP4* overexpressing cells, but rather had a synergistic effect in causing cell death (Fig. 7e). Conversely, *CASP4* overexpression did not induce death in cells treated with anticancer drugs that cause apoptosis, such as cisplatin and bleomycin (Fig. S7b). Furthermore, *CASP4* KD significantly suppressed the reduction in cell viability induced by high-dose IFN-γ (Fig. S7c). Next, we examined which type of cell death is caused by IFN-γ. IFN-γ is known to cause ferroptosis in tumor cells[27]. IFN-γ treatment was found to induce an increase in lipid peroxide, an indicator of ferroptosis, but there was no significant difference between mock and *CASP4* overexpressing cells in this respect (Fig. S7d and S7e). In addition, synergistic cell death was induced in *CASP4* overexpressing cells even under treatment with ferrostatin-1, a ferroptosis inhibitor (Fig. S7f). On the other hand, high-dose IFN-γ treatment induced an increase in cleaved Gasdermin D (GSDMD) which plays a major role in pyroptosis (Fig. 7f). These results indicate that the main mode of cell death occurring in *CASP4* overexpressing cells is pyroptosis mediated by GSDMD cleavage.

## Discussion

Previous studies have demonstrated that CASP4 expression in lung cancer cells plays a role in malignant behavior[16–18]. The present study investigated the underlying mechanisms in further detail. Based on our clinical data and TCGA dataset, we confirmed that high *CASP4* expression in lung adenocarcinoma cells was a poor prognostic factor. Notably, higher *CASP4* level was associated with shorter OS in patients with advanced-stage lung adenocarcinoma (stages III–IV).

In general, CASP4 induces cell death[28–30]. However, our study demonstrated that CASP4 expression alone barely induced death in lung adenocarcinoma cells, which can be attributed to the expression of antiapoptotic genes, such as *BCL2A1* and *BCL2L1*. Further, CASP4 activates NF-κB, which may be involved in the induction of antiapoptotic gene expression by CASP4. Moreover, antiapoptotic BCL-2 family members are the key factors for suppressing apoptosis by inhibiting the caspase cascade, conferring resistance to chemotherapy in tumors[31]. CASP4 expression in cancer cells may be involved in the acquisition of anticancer drug resistance.

The present study revealed that CASP4 induces enhanced angiogenesis and cell migration in lung adenocarcinoma cells. Fan et al. reported that CASP4/Casp11 regulates angiogenesis in vascular endothelial cells by inhibiting γ-secretase activity and Notch-1 signaling[20]. However, altered expression of *NOTCH1* and its downstream transcription factors *HEY1*, *HEY2*, and *HES1* was not observed in *CASP4* overexpressing or -KO lung cancer cells (Fig. S2e). Instead, we observed that CASP4 induced PTGS2 and EPHA2 expression, which is mainly located downstream from NF-κB. Studies have shown that PTGS2 and EPHA2 enhance cell motility, metastatic potential, and angiogenesis in cancer cells[32–35]. Given that the restoration of PTGS2 and EPHA2 in *CASP4* knockout cells led to improved cell motility, we hypothesize that CASP4 enhances the invasive and metastatic capabilities of tumor cells by modulating the expression of these genes.

NF-κB plays a crucial role in tumor progression by targeting genes involved in the growth, survival, metastasis, invasion, angiogenesis, and chemo- and radiotherapy resistance of tumor cells[36]. Constitutive or aberrant activation of NF-κB is frequently observed in lung cancer[37,38]. Starczynowski et al. reported that TNF receptor-associated factor 6 (TRAF6) induces continuous NF-κB activation in human lung cancer[39]. In addition, CASP4 interacts with TNF receptor-associated factor 6 and mediates NF-κB activation in response to LPS and TNF-α in macrophages[40,41]. In the present study, *CASP4* expression alone without LPS or TNF-α treatment was found to activate NF-κB in lung adenocarcinoma cells. Further studies are required to elucidate the association of CASP4 with TNF receptor-associated factor 6.

When syngeneic and allogeneic tumors were transplanted into mice, *Casp11* KO cells showed decreased tumor growth, lung metastasis, and angiogenesis both in immunocompetent mice and immunodeficient mice, suggesting that these phenotypic differences occur regardless of host tumor

microimmunity. However, CASP4 is intricately involved in innate immunity, and future studies should examine the relation between CASP4 and the tumor microenvironment.

We demonstrated that CASP4 expression is considerably upregulated in lung adenocarcinoma cells through activation of STAT1 signaling. The tumor microenvironment is characterized by ER stress conditions, such as hypoxia and glucose starvation[42,43]. Further, STAT1 is activated by ER stress[44] and is induced by various factors such as IFN-α, IFN-β, and IFN-γ, which are further induced by several immune cells, such as CD4$^+$ T cells, CD8$^+$ T cells, and natural killer cells, in the tumor microenvironment[45]. Our result that CASP4 is induced by ER stress or IFN-γ may indicate CASP4 expression in lung adenocarcinoma cells and reflect the state of the microenvironment at the tumor cell periphery. As the association between CASP4 and tumor microenvironment was not explored in this study, further investigation is warranted.

IFN-γ is a pleiotropic cytokine in tumor development depending on its concentration. Low IFN-γ concentrations were reported to promote tumor progression, enhancing the stemness of lung cancer cells[46]. Low IFN-γ levels are maintained to sustain NF-κB signaling activation in the tumor microenvironment, simultaneously inducing angiogenic growth and antiapoptotic factors, rendering immune checkpoint inhibitors ineffective[47]. Therefore, intervention to change the NF-κB signature in the tumor microenvironment can obtain a favorable therapeutic effect of immune checkpoint inhibitors. Conversely, high IFN-γ levels directly suppress tumor growth and induce cell death[48,49], but the underlying mechanism remains unknown. Wang et al. reported that IFN-γ inhibits cysteine uptake by tumor cells and promotes ferroptosis in tumor cells[27]. On the other hand, IFN-γ has also been reported to induce pyroptosis in corneal epithelial cells via the JAK2/STAT1 pathway[50]. In the present study, high level of IFN-γ was found to induce pyroptosis by converting GSDMD to its activated form in lung adenocarcinoma cells, which was enhanced by CASP4 expression. Furthermore, IFN-γ-induced cell death was promoted by fludarabine, a STAT1 inhibitor. The synergistic effect on cell death is likely attributable to the suppression of STAT1 signal-dependent cell survival signals by fludarabine. Further studies are required for indepth characterization of the signal pathway.

In conclusion, we demonstrated that CASP4 is a promoter of tumor angiogenesis and metastasis and facilitates IFN-γ-mediated pyroptosis. Suppression of CASP4 or IFN-γ therapy are potential therapeutic options for patients with lung adenocarcinoma and high CASP4 expression.

## Methods
### Human samples
This study included 77 patients with NSCLC including 56 consecutive patients with lung adenocarcinoma who underwent surgical resection for lung cancer at the Aichi Cancer Center Hospital, Nagoya, Japan, between May 2019 and June 2020. Patient characteristics and selection flowchart are shown in Table S1 and Fig. S1a.

### The Cancer Genome Atlas (TCGA) database analysis
Human lung adenocarcinoma data were derived from TCGA Research Network (http://cancergenome.nih.gov/). The dataset from this resource that supports the present study findings is available on cBioPortal (https://www.cbioportalorg/)[51,52].

### Cell culture
NCI-H292 (American Type Culture Collection, USA), A549 (Japanese Cancer Research Bank, Japan), H1975 (American Type Culture Collection, USA), and Lewis Lung Carcinoma (3LL; Japanese Cancer Research Bank, Japan) cell lines were cultured in RPMI-1640 medium (Wako, Japan) supplemented with 10% fetal bovine serum (FBS) (Thermo Fisher Scientific, USA), 2 mM L-glutamine (Wako, Japan), and 1% penicillin/streptomycin (Wako, Japan) at 37 °C in a 5% $CO_2$ atmosphere. Myco-Lumi Luminescent Mycoplasma Detection Kit (Beyotime, China) was routinely used to detect the presence of mycoplasma infection.

## Establishment of CASP4, PTGS2, EPHA2 overexpressing cell lines using lentivirus

A lentiviral delivery system was used to establish A549 and NCI-H292 cells with stable *CASP4, PTGS2, EPHA2* overexpression. The human *CASP4, PTGS2, EPHA2* open reading frame was cloned into pLenti-CMV-MCS-GFP-SV-puro (Addgene #73582) plasmid. Catalytically inactive CASP4 (C258A) variants were generated via QuikChange site-directed mutagenesis following the manufacturer's protocol (Agilent, USA). The primers used in this process were as follows: CASP4(C258A) (F:5′-ATCATTGTC-CAGGCCGCCAGAGGTGCAAACC-3′; R:5′-GGTTTGCACCTCTGGCGGCCTGGACAATGAT-3′). Lentiviruses were packaged into 293 T cells by transfecting these cells with *CASP4, PTGS2, EPHA2* overexpressing plasmids, psPAX2 plasmid (lentiviral packaging, Addgene #12260), and pCMV-VSV-G plasmid (envelope, Addgene #8454) at a molar ratio of 1:1:1 using the transfection reagent FuGENE HD (Promega, USA). The virus particles were collected after 24 h. Target cells were infected with the virus for 24 h and selected using 2 µg/mL puromycin (Wako, Japan) for approximately 1 week to establish stable cell lines before being used for experiments. The cells infected with the empty vector (mock) were used as controls in experiments evaluating *CASP4, PTGS2, EPHA2* overexpressing cells.

## Establishment of CASP4 knockout (KO) cell lines via CRISPR/Cas9

We designed a pair of single guide RNA sequences targeting the exon 2 of *CASP4* (F: 5′-CAAGAGAAGCAACGTATGGC-3′; R: 5′-GCCA-TACGTTGCTTCTCTTGC-3′) to generate the *CASP4* KO NCI-H292 cell line. Sequence analysis confirmed the genetic deletion in exon 2 of human CASP4 and confirmed that translation ends at 75 amino acids (molecular weight = 8.772 kDa) and 79 amino acids (molecular weight = 9.436 kDa) from these mutant sequences (Fig. 2c). The levels of CASP4 were confirmed using immunofluorescence staining and immunoblotting (Fig. 2a, b). To establish the mouse 3LL cell line, the single guide RNA sequences targeting the exon 2 of *Casp11* (F: 5′-CACCGCATATGGC-3′; R: 5′-AAACGCTCTTGC-3′) were used. These single guide RNAs were cloned into a lentiCRISPRv2neo vector obtained from Addgene (plasmid #98292). *CASP4*- and *Casp11*-KO cells and empty vector control cells (mock) were established using the previously described lentiviral infection method and selected using 500 µg/mL G418 (Wako, Japan) for approximately 10 days.

## CASP4 or signal transducer and activator of transcription 1 (STAT1) knockdown (KD) via small hairpin RNA (shRNA)

shRNAs targeting *CASP4* and *STAT1* to the NCI-H292, A549, H1975 cells are shown in Table S2. These sequences were used to generate the lentiviral vector pLKO.1 puro (Addgene #8453)-based shRNAs targeting *CASP4* or *STAT1*. Cells infected with the empty vector (mock) were used as controls in experiments evaluating *CASP4* or *STAT1* KD cells.

## Wound healing assay

Wound healing assay was used to assess the migration ability of lung cancer cell lines. NCI-H292 and A549 cells were seeded into 6-well plates and the center of the well was scraped using 200-µL pipette tip. After washing with PBS, the cells were cultured in RPMI-1640 medium supplemented with 10% FBS and 1% penicillin and streptomycin. Images were captured after 18 or 24 h.

## Transwell migration assay

Transwell assay inserts (8.0 µm pore-size membranes) (BD Falcon, Switzerland) were placed in 24-well plates. Cells ($1 \times 10^4$) resuspended in FBS-free RPMI-1640 medium were seeded into the upper chamber, whereas RPMI-1640 along with 10% FBS was added to the lower chamber. After incubation for 24 h, the cells were fixed with 4% paraformaldehyde and permeabilized with 0.4% Triton X in PBS. Non-invading cells were mechanically wiped using cotton swabs and visualized by Giemsa staining.

Images were obtained using a BZX-800 fluorescence microscope (Keyence, Japan). Quantitative analysis of migrated cells was performed using ImageJ software (National Institutes of Health, USA).

## Cell viability and cytotoxicity assays

Cells were seeded into a 96-well plate at a density of $1 \times 10^3$ cells/well and treated as indicated. Cell Counting Kit 8 (CCK8) (Dojindo, Japan) was used to measure cell viability, and Cytotoxicity LDH Assay kit-WST (Dojindo, Japan) was used to collect supernatants for cytotoxicity analysis, following the manufacturer's instructions. The absorbance was measured at 450 nm using a microplate reader (iMark, Bio-Rad Laboratories, USA).

## RNA extraction and reverse transcription quantitative polymerase chain reaction (RT–qPCR)

RNA isolation was performed using Sepasol-RNA I Super G (Nacalai, Japan), following the manufacturer's instructions. Total RNA was synthesized using ReverTra Ace qPCR RT kit (Toyobo, Japan). RT–qPCR was performed using THUNDERBIRD SYBR qPCR Mix (Toyobo, Japan) and Thermal Cycler Dice Real Time System Lite (Takara, Japan). Glyceraldehyde-3-phosphate dehydrogenase was used as a control to evaluate loading and PCR efficiency. Table S2 lists the primers used for RT–qPCR.

## Western blot assay

Total cell lysates were extracted using lysis buffer containing 1% Triton X-100 and complete protease inhibitor cocktail (Roche Diagnostics, Germany). The protein concentration of cell lysates was determined using Bio-Rad Quick Start Bradford (Bio-Rad, USA). Cellular proteins were separated by 10% sodium dodecyl sulfate–polyacrylamide gel electrophoresis and transferred to polyvinylidene fluoride membranes for immunoblotting. After blocking with 5% skim milk for 1 h at 20 °C, the membranes were incubated overnight with primary antibody (1:1000 dilution) at 4 °C. After washing, the membranes were labeled with the secondary antibody (1:5000 dilution) for 1 h at 20 °C. Immunodetection was performed using ECL Prime with the ImageQuant LAS 4000 mini system (GE Healthcare, USA), following the manufacturer's protocol. Table S2 lists the antibodies used in this study.

## Histopathology and immunohistochemistry

Formaldehyde-fixed lung tissue specimen was paraffin-embedded, sectioned at a thickness of 4 µm, and stained with hematoxylin and eosin. Immunostaining was performed using anti-CD31 monoclonal antibody (1:200 dilution, Table S2). After washing with PBS, the sections with primary antibodies were immunohistochemically stained with Histofine Simple Stain MAX PO (Nichirei Bioscience, Japan). The sections were washed again to remove primary antibodies, stained with dimethylaminoazobenzene for approximately 5 min, and then counterstained with Mayer's hematoxylin. A board-certified pathologist examined the sections for histological changes. For CD31 quantification, the immunohistochemistry staining intensity was calculated as the percentage of CD31-positive area per field of view using ImageJ software.

## Immunofluorescence staining

For immunolabeling studies, cells were seeded overnight in cover glass and fixed with 4% paraformaldehyde for 10 min. Additionally, 0.1% Triton X-100 was used to permeabilize the cells, and 1% bovine serum albumin was added to block nonspecific binding for 1 h. Cells were stained with primary monoclonal antibody for 1 h and then stained with fluorescent secondary antibody for 30 min. The nuclei were counterstained with DAPI. Confocal images were obtained using a fluorescence microscope (BZX-800, Keyence, Japan). Cytoplasmic / nuclear localization ratio for p65 was performed by counting 10 cells per field using ImageJ software. Table S2 shows the antibody reagents used in this study.

## Flow cytometry analysis

A total of $5 \times 10^5$ cells were incubated with the indicated antibodies for 30 min at 4 °C. All cells were washed with PBS twice and diluted with fluorescence-activated cell sorting buffer (2% FBS in PBS). After treatment, they were analyzed using fluorescence-activated cell sorting Calibur system and CellQuest Pro software (BD Biosciences, USA). For detection of lipid peroxidation, cells were incubated in the C11 BODIPY 581/591 (Dojindo, Japan) diluted in serum-free medium to a final concentration of 2 μM for 30 min in dark. Table S2 lists the antibodies used in this study.

## Annexin-V/PI double staining assay

Cells were treated with IFN-γ at the indicated concentrations for 48 h. Then, they were harvested (including attached and detached cells) and washed and resuspended with PBS. Dying cells were assessed using Annexin-V-PE (MBL, Japan) and propidium iodide (Dojindo, Japan), according to the manufacturer's protocol.

## RNA sequencing

RNeasy Mini kit (cat. no, 2301158, Qiagen) was used to extract total RNA from NCI-H292 cells that were mock-infected with the empty vector (n = 3), *CASP4* overexpressing NCI-H292 cells (n = 3), and *CASP4* KO NCI-H292 cells (n = 3), following the manufacturer's instructions. TruSeq standard mRNA library kit (cat. no, 20020594; Illumina) was used to prepare the libraries, and NovaSeq 6000 (Illumina) was used to sequence them. TopHat2 version 2.1.0 was used to map the sequence reads to the human reference genome HG19. Cufflinks version 2.2.1 was used to assemble the transcripts. The Cuffdiff program was used to identify differentially expressed genes. Metascape.30 was used to perform gene ontology (GO) analysis using CDS expression data from RNA sequencing[53]. Transcription factors that regulate the enriched genes were searched using Transcriptional Regulatory Relationships Unraveled by Sentence-based Text mining (TRRUST)[22].

## Luciferase reporter assay for measuring NF-κB activity

A total of $4 \times 10^4$ cells were seeded into 24-well plates and cotransfected with pGL4.32 [luc2P/NF-kB-RE/Hygro] (Promega, USA) and pRL-TK (Promega, USA) vectors for normalization (total plasmid amount: 0.5 μg) using FuGENE HD (Promega, USA). After 24 h, the cells were lysed with 100 μL of passive lysis buffer. The luciferase activity was measured using the dual-luciferase reporter assay system (Promega, USA).

## Animals

Male C57/BL6L mice (age: 6 weeks) and male BALB/c nu/nu mice (age: 6 weeks) were purchased from CLEA Japan (Tokyo, Japan). The mice were allowed to acclimatize for at least 7 days before use. The mice were housed in a controlled laboratory environment with standard temperature and light/dark cycles at the University of Occupational and Environmental Health. We have complied with all relevant ethical regulations for animal use.

## Subcutaneous tumor mouse model

The mice were anesthetized with 4 mg/kg midazolam (Wako, Japan), 0.3 mg/kg medetomidine (Wako, Japan), and 5 mg/kg butorphanol (Wako, Japan). Then, $3 \times 10^5$ mock-infected or *Casp11*-KO 3LL cells in 100 μL of PBS were subcutaneously injected using a 26-gauge needle (n = 5–6). The volume of xenografts was measured twice weekly for 28 days in C57/BL6L mice and for 24 days for BALB/c nu/nu mice, and calculated using the following formula: $L \times W^2 \times 0.5$. C57/BL6L mice and BALB/c nu/nu mice were sacrificed 28 days and 24 days after subcutaneous injection, respectively, and tumor size and weights were compared between the two groups.

## Statistics and reproducibility

Statistical analysis was performed using Prism software version 8 (Graph-Pad, USA). Two-tailed Student's *t* test was used to assess differences between two groups. The log-rank and Gehan–Breslow–Wilcoxon test were used for Kaplan–Meier analysis. All in vitro experiments were performed at least thrice and the data were shown as mean ± standard error of the mean; $p < 0.05$ were considered indicative of statistical significance.

## Study approval

All human studies received ethical approval of the Institutional Review Board of Aichi Cancer Center (No. 2018–2–20) and the Ethics Committee of Medical Research, University of Occupational and Environmental Health, Japan (No. UOEHCRB20-015) and conducted following the guidelines. Appropriate informed consent was obtained from all human participants. The study protocol complied with the principles of the Declaration of Helsinki. All animal studies were performed after obtaining approval from the Ethics Committee of the Laboratory Animal Care and Use Committee at the University of Occupational and Environmental Health (Approval Number: AE21-007).

## Data availability

All RNA sequencing library data from NCI-H292 cells are available through the Gene Expression Omnibus database (https://www.ncbi.nlm.nih.gov/geo/) under accession code GSE235296. Source data underlying main figures are presented in Supplementary data 1-7. The graphical abstract representing the overview of this research is shown in Fig. S8. FACS gating strategy corresponding to Fig. 7c is presented in Figure S9. Full western blot images are included in Fig. S10 – Fig. S12. All other data are available from the corresponding author (or other sources, as applicable) on reasonable request.

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

## Acknowledgements

This work was supported by the Scientific Research Fund of the Ministry of Education, Culture, Sports, Science and Technology (MEXT) of Japan (grant 23K07641 to M.E., 21K07972 to T.D., and 22K08296 to K.O.) and UOEH Research Grant for Promotion of Occupational Health (2021).

## Author contributions

Y.C. designed the study, performed and analyzed most experiments, and wrote the paper. T.D. designed and supervised the study and wrote the paper. K.O., K.S., S.N., and K.-Y.-W. performed experiments. K.Y., H.M., K.M., and H.K. managed the data and developed the clinical data set. K.Y. and M.E. coordinated, designed, and supervised the study as well as wrote the paper.

## Competing interests

The authors declare no competing Interests.
