## [Peer Review File · Communications Biology]

Reviewers' comments:

Reviewer #1 (Remarks to the Author):

The authors investigated the expression levels of CASP4 mRNA in a total of 77 patients with NSCLC and the overall survival probability in CASP4 high and CASP4 low patients in stage I/II and stage III/IV. The data reported indicate that patients with advanced cancer and higher CASP4 levels had shorter OS. Next, the authors, to investigate the role of CASP4 in NSCLC, established CASP4-overexpressing cells lines and performed RNA sequencing and gene enrichment analyses of mock and CASP4 transfected NCI-H292 cells. Moreover, the authors validated by qPCR the upregulation of some genes associated to both angiogenesis and positive regulation of cell migration in the CASP4 overexpressing cell line NCI-H292. RNA seq and gene expression analysis were also performed in NCI-H292 mock and CASP4 knockdown cells. Overall, the data show a correlation of CASP4 expression levels with the expression of genes associated to angiogenesis and migration in NSCLC cell lines. Functional cell migration assays and studies in syngeneic mouse models murine models (tumor growth and metastasis) nicely show the relevance of CASP4/11 in relevant biological and tumor processes.

The authors also identified the NF- κ B pathway as a mediator of the gene expression variations observed in cell lines with modulation of CASP4 and investigated the effects of IFN γ and ER stress on CASP4 expression cell death and survival.

The paper is interesting and original. The data reported stimulate further investigations on the role of CASP4 in tumor progression and on its regulation. Signaling pathways and functional data need to be improved and analyzed quantitatively. Interpretation of data shown in Fig. 5-7 should be less assertive. Discussion could be more focused on the data presented and less scattered.

Specific comments/requests

Overall, the presentation of data on protein expression should be improved. The western blots with the CASP4 antibody sometimes show 1 band other times 2 or 3 bands. Band signals should be quantified in all blots. Table S2 show all primers used in the study I do not see the list and features of the antibodies used that should be included.

Fig. 1B add molecular weight size in the blot, quantification and comments on the bands detected.

Fig. 1D add p-values in the caption.

Fig. 1G add a description of the BCL2/BCL2A1/BCL2L1 e MCL1 below the histograms (Negative regulation of apoptosis)

Fig. S1D add molecular weight size in the blots and include band signals quantification, you have three bands please describe/comment the multiple bands observed in your analysis.

Fig. 2A add molecular weight size in the blot and comments on the bands detected and the increase of signal in the lower band.

Fig. 2C add p-values in the caption.

Fig. 4/5/6 add molecular weight size in the blot and band signal quantifications.

Fig. S4B and S5B add band signals quantification and molecular weight size.

Fig. S4C/S4D show quantitative data (number of images counted, number of cells with p65 nuclear localization).

Upregulation of genes associated with cell migration and angiogenesis by CASP4 is mediated through the NF- κ B pathway in NSCLC cells.

I suggest to change the title of the above paragraph i.e. "NF- κ B pathway contributes to the upregulation of genes... because other pathways could be implicated in the gene regulation.

I think that Fig. 5A and Fig. 5B are mislabeled. In A) is C4oE vs mock?

Please clarify the abbreviation C4KD in the text.

Sentence lines 35-307 pag. 19 "These results indicate that CASP4 promotes cell migration and angiogenesis through NF- κ B". I suggest revising it for example "NF- κ B contributes to"

Please indicate the time of treatment in panels Fig. 6C and D. Please also show the concentration of IFN γ used in the western blot analysis (Fig. 6D) and quantify the band signals, also comment the multiple bands detected.

Experiments with IFN γ . I understand that the data shown in Fig. 7C and 7D are representative of multiple experiments. The increase of apoptosis in C4OE is small (22.9 vs 16%) therefore a Table with all your data and a statistical analysis is required.

Reviewer #2 (Remarks to the Author):

The authors studied the role of CASP4 (casp11 in mice) in non-small cell lung cancer (NSCLC) by generating multiple gain-of-function or loss-of-function mutant NSCLC cell lines. They reported that Casp4 promotes the progression, migration and metastasis of NSCLC tumors via NF κ B mediated induction of genes associated with cell migration and angiogenesis. They further reported that Casp4 upregulation in NSCLC cells was caused by ER stress and IFN γ , which increased the sensitivity of NSCLC cells to high dose IFN γ induced cell death. Although high levels of casp4 has been previously shown to be associated with poor prognosis in NSCLC, this paper add incremental values into the function of CASP4 in promoting non-small cell lung cancer. The experiments are well-designed, and data are well presented. However, most of the molecular mechanisms the author identified are based on seq-data or bioinformatic analysis and lack experimental evidence, as such I have some specific comments as follows

1. In Fig S1 and S2, the rationale that the authors chose those specific genes to present is not clear because some of these genes are obviously not among the top DEGs, such as KLF4, myadm and igfbp5. Are those genes selected because they are the only overlapping DEGs in both the overexpression and knockout gene sets? If that was the case, the author should list all the overlapping DEGs in these two data sets (mock vs C4OE and mock vs C4KO);

2. Fig 2D: the y axis should be C4KO rather than C4OE; Have the author tested and confirmed these six selected genes by Q-PCR in C4OE and C4KO cells as did for C4KD cells in FigS2C? how about the expression of these genes in mutated H1975 cells?

3. Fig 3: Since NFKb inhibitor diminished the induction of PTGS2, SERPINE1 and EPHA2 in C4OE cells, does it also reduce the migration of these cells in the wound healing assay or transwell experiment?

4. It would be greatly improved if the author experimentally tested one of these genes in contributing to the enhanced ability of migration or angiogenesis of NSCLC cells.

4. Fig 4 very clearly shows that C11KO reduced the ability of tumor progression and metastasis in vivo. However, this was done in an immunocompetent mouse strain, suggesting a role of immune system in contributing to the observed difference rather than the changes in angiogenesis. The reduced CD31 expression may be simply due to the markedly reduced tumor size in C11KO tumors. How about tumor growth and metastasis of C11KO vs Mock cells in immune deficient mice? The potential contribution Casp4 in shaping tumor immune microenvironment should be at least discussed.

5. Fig 5G, NFKb inhibition also significantly reduced the expression of CASP4 in C4OE cells, suggesting that CASP4 expression is partially driven by activated NFKb signaling. Hence, the decreased expression of PTGS2 and SERPINE1 upon NFKB inhibitor treatment may be a result of reduced expression of CASP4. This need to be clarified.

6. In Fig7B, the usage of C4OE C258A cells should be more clearly clarified. Are the cell dead upon IFNg treatment in a form of proptosis? or Ferroptosis? How about the role of Gasdermin D in this process?

6. Fig 6C shows that low dose of IFNg promoted the expression of CASP4, is that also the case for high dose IFNg treatment (the dosage used in killing assay?). It is a little confused that IFNg can both promote CASP4 expression and induce cell death in CASP4 overexpressing cells. The author should compare casp4 expression in mock and C4OE cells upon high dose IFNg treatment. Moreover, does STAT1 inhibition decrease the sensitivity of C4OE cells to IFNg induced death?

Reviewer #3 (Remarks to the Author):

The title of the manuscript "Caspase-4 promotes both metastasis and interferon- γ -induced cell death in non-small cell lung cancer" promises more than it delivers.

While the paper has novel observations, and the role of caspase-4 in non-small cell subtype adenocarcinoma is well investigated, the presentation of the findings makes it extremely difficult to follow the significance of the manuscript. In its present form, the manuscript can only be appreciated by researchers who work on intracellular signaling in depth. Although the methodology is not particularly novel it is nicely performed.

I believe the manuscript would benefit from rethinking the purpose of the study and reanalysis of some data.

1. The manuscript does not investigate the role of caspase-4 in NSCLC in general, but NSCLC-adenocarcinoma.

2. While they collected 77 NSCLC patient samples for the study which I think is a fantastic opportunity to make it clinically relevant, the authors included all the patients. At the same time, some of them were affected by squamous cell carcinoma that was not studied separately.
3. I would expect a figure for patient selection criteria, to focus the study on adenocarcinoma patients
4. I think this would be particularly important, as all the in vitro methodology was performed in lung adenocarcinoma cell lines.
5. I recommend a more detailed, but simple summary figure, to make the results more widely understandable.
6. A more straightforward conclusion is needed

Reviewer #1 (Remarks to the Author):

The authors investigated the expression levels of CASP4 mRNA in a total of 77 patients with NSCLC and the overall survival probability in CASP4 high and CASP4 low patients in stage I/II and stage III/IV. The data reported indicate that patients with advanced cancer and higher CASP4 levels had shorter OS. Next, the authors, to investigate the role of CASP4 in NSCLC, established CASP4-overexpressing cells lines and performed RNA sequencing and gene enrichment analyses of mock and CASP4 transfected NCI-H292 cells. Moreover, the authors validated by qPCR the upregulation of some genes associated to both angiogenesis and positive regulation of cell migration in the CASP4 overexpressing cell line NCI-H292. RNA seq and gene expression analysis were also performed in NCI-H292 mock and CASP4 knockdown cells. Overall, the data show a correlation of CASP4 expression levels with the expression of genes associated to angiogenesis and migration in NSCLC cell lines. Functional cell migration assays and studies in syngeneic mouse models murine models (tumor growth and metastasis) nicely show the relevance of CASP4/11 in relevant biological and tumor processes.

The authors also identified the NF- κ B pathway as a mediator of the gene expression variations observed in cell lines with modulation of CASP4 and investigated the effects of IFN γ and ER stress on CASP4 expression cell death and survival.

The paper is interesting and original. The data reported stimulate further investigations on the role of CASP4 in tumor progression and on its regulation. Signaling pathways and functional data need to be improved and analyzed quantitatively. Interpretation of data shown in Fig. 5–7 should be less assertive. Discussion could be more focused on the data presented and less scattered.

Thank you very much for your positive comments. Signaling pathways and functional data have all been quantified based on the specific comments below. In addition, we have revised the discussion to make it more focused.

Specific comments/requests

1. Overall, the presentation of data on protein expression should be improved. The western blots with the CASP4 antibody sometimes show 1 band other times 2 or 3 bands. Band signals should be quantified in all blots. Table S2 show all primers used in the study I do not see the list and features of the antibodies used that should be included.

Thank you for pointing this out. All protein levels have been quantified in the revised manuscript. In addition, the antibodies used in this study are shown in Table S2.

2. *Fig. 1B add molecular weight size in the blot, quantification and comments on the bands detected.*

We have shown the molecular weight and CASP4 main band in Figure 1B; the CASP4 level is also quantified.

3. *Fig. 1D add p-values in the caption.*

We added the following sentence in the Figure 1D legend:

“The horizontal axis shows accumulative hypergeometric p-values.”

4. *Fig. 1G add a description of the BCL2/BCL2A1/BCL2L1 MCL1 below the histograms (Negative regulation of apoptosis)*

We have added the recommended description in Fig. 1F (Fig. 1G before revision).

5. *Fig. S1D add molecular weight size in the blots and include band signals quantification, you have three bands please describe/comment the multiple bands observed in your analysis.*

We have added arrows to the molecular weight and CASP4 body bands in Figure S1F (Fig. S1D before revision) and quantified the CASP4 level. Additionally, the notation N.S. has been added for nonspecific bands.

6. *Fig. 2A add molecular weight size in the blot and comments on the bands detected and the increase of signal in the lower band.*

The molecular weight and CASP4 signal band have been added to Figure 2A.

We also performed additional immunostaining of *CASP4* KO cells and confirmed the disappearance of CASP4 intracellular signals (Fig. 2B). We confirmed *CASP4* gene mutations in two alleles through genotyping and confirmed that translation ends at 75 amino acids (molecular weight = 8.772 kDa) and 79 amino acids (molecular weight = 9.436 kDa) from these mutant sequences (Fig. 2C).

We have added information on how to check *CASP4* KO cells. These findings have been reported on page 7, lines 89 - 92 of the revised manuscript.

7. *Fig. 2C add p-values in the caption.*

We have added the following sentence to the legend of Figure 2E (Figure 2C before revision):

“The horizontal axis shows accumulative hypergeometric p-values.”

8. *Fig. 4/5/6 add molecular weight size in the blot and band signal quantifications.*

In the revised manuscript, we have added molecular weight and band signal quantification to each immunoblot image.

9. *Fig. S4B and S5B add band signals quantification and molecular weight size.*

We have added molecular weight and band signal quantification to each immunoblot image in Figure S5B/S6B (Figure S4B/S5B before revision).

10. *Fig. S4C/S4D show quantitative data (number of images counted, number of cells with p65 nuclear localization).*

We have added a bar graph of the quantification of nuclear/cytoplasmic ratio of p65 intensity in Figure S5C/D (Figure S4C/D before revision) (n = 10).

11. *Upregulation of genes associated with cell migration and angiogenesis by CASP4 is mediated through the NF- κ B pathway in NSCLC cells. I suggest to change the title of the above paragraph i.e. “NF- κ B pathway contributes to the upregulation of genes... because other pathways could be implicated in the gene regulation.*

Thank you for the insightful suggestion. We have modified the title as suggested (page 39, lines 640 - 641 of the revised manuscript).

12. *I think that Fig. 5A and Fig. 5B are mislabeled. In A) is C4oE vs mock?*

Please clarify the abbreviation C4KD in the text.

Thank you for pointing this out. Fig. 5A has been revised to C4OE vs mock. In addition, the full form of KD (knockdown) is mentioned on page 7, lines 100 of the revised manuscript.

13. *Sentence lanes 35-307 pag. 19 “These results indicate that CASP4 promotes cell migration and angiogenesis through NF- κ B”. I suggest revising it for example “NF- κ B contributes to”*

We have modified this sentence as suggested (page 20, lines 323-324 of the revised manuscript).

14. Please indicate the time of treatment in panels Fig. 6C and D. Please also show the concentration of IFN γ used in the western blot analysis (Fig. 6D) and quantify the band signals, also comment the multiple bands detected.

Thank you for your important comments. We have added the time of treatment in Figure 6C and D and legends. The IFN- γ treatment concentration has been added to the Figure 6D and legend. We have also quantified CASP4 signal band and added “N.S” for nonspecific band.

15. Experiments with IFN γ . I understand that the data shown in Fig. 7C and D are representative of multiple experiments. The increase of apoptosis in C4OE is small (22.9 vs 16%) therefore a Table with all your data and a statistical analysis is required.

We have added graphs showing quantitative analysis in Fig. 7C and D.

Reviewer #2 (Remarks to the Author):

The authors studied the role of CAPS4 (casp11 in mice) in non-small cell lung cancer (NSCLC) by generating multiple gain-of-function or loss-of-function mutant NSCLC cell lines. They reported that Casp4 promotes the progression, migration and metastasis of NSCLC tumors via NFkB mediated induction of genes associated with cell migration and angiogenesis. They further reported that Casp4 upregulation in NSCLC cells was caused by ER stress and IFNg, which increased the sensitivity of NSCLC cells to high-dose IFNg induced cell death. Although high levels of casp4 has been previously shown to be associated with poor prognosis in NSCLC, this paper add incremental values into the function of CASP4 in promoting non-small cell lung cancer. The experiments are well-designed, and data are well presented. However, most of the molecular mechanisms the author identified are based on seq-data or bioinformatic analysis and lack experimental evidence, as such I have some specific comments as follows

Thank you for your very important comment. As suggested, we have added some new experiments in the study.

1. In Fig S1 and 2, the rationale that the authors chose those specific genes to present is not clear because some of these genes are obviously not among the top DEGs, such as KLF4, myadm and igfbp5. Are those genes selected because they are the only overlapping DEGs in both the overexpression and knockout gene sets? If that was the case, the author should list all the overlapping DEGs in these two data sets (mock vs. C4OE and mock vs. C4KO).

Yes, we selected overlapping DEGs for both overexpression and knockout gene sets. Overlapping DEGs have been indicated in Fig. 2G. The relevant description is presented in page 17, lines 266 - 268 in the revised manuscript. In addition, Fig. S1H and S2C have been changed to focus on *EPHA2* and *PTGS2*, for which significant differences were observed with reproducibility in Q-PCR.

2. Fig 2D: the y axis should be C4KO rather than C4OE; Have the author tested and confirmed these six selected genes by Q-PCR in C4OE and C4KO cells as did for C4KD cells in FigS2C? how about the expression of these genes in mutated H1975 cells?

Thank you for the valuable inputs. We have corrected Fig.2F (Fig. 2D before revision). In C4OE and C4KO, six selected genes were also confirmed by Q-PCR, and similar to RNA-seq FPKM, significant differences were observed in *EPHA2* and *PTGS2*. Based on this, we focused on the *EPHA2* and *PTGS2* genes in Fig. 2H and I (Fig. 1F, 2G before revision) and Fig. S1H and S2C (S1F,

and S2C before revision). We attempted to create CASP4 overexpressing cells using H1975 cells but were unsuccessful in achieving CASP4 overexpression. The data for H1975 knockdown (KD) are presented in Fig. S2. These findings have been reported on page 17, lines 268 - 270 of the revised manuscript.

3. Fig 3: Since NFKb inhibitor diminished the induction of PTGS2, SERPINE1 and EPHA2 in C4OE cells, does it also reduce the migration of these cells in the wound healing assay or transwell experiment?

When treated with NF-kB inhibitor (BAY11-7082), a decrease in cell migration was observed in both mock and C4OE cells in wound healing assays. These data have been added to Fig 5I and J. These findings have been reported on page 20, lines 322 - 324 of the revised manuscript.

4. It would be greatly improved if the author experimentally tested one of these genes in contributing to the enhanced ability of migration or angiogenesis of NSCLC cells.

We performed experiments to rescue *PTGS2* and *EPHA2*, whose expression was downregulated by *CASP4* KO. Wound healing assay confirmed the restoration of the migration ability of *CASP4* KO cells after rescuing *PTGS2* or *EPHA2* (Fig. 3G–3J and S3E and F). These findings have been reported on page 17, lines 281 - 284 of the revised manuscript.

5. Fig 4 very clearly shows that C11KO reduced the ability of tumor progression and metastasis in vivo. However, this was done in an immunocompetent mouse strain, suggesting a role of immune system in contributing to the observed difference rather than the changes in angiogenesis. The reduced CD31 expression may be simply due to the markedly reduced tumor size in C11KO tumors. How about tumor growth and metastasis of C11KO vs. mock cells in immune-deficient mice? The potential contribution Casp4 in shaping tumor immune microenvironment should be at least discussed.

Based on your suggestion, we also performed subcutaneous tumor transplantation on C11KO vs. mock in immune-deficient mice (BALB/c nu/nu). Similar to immunocompetent mice, differences in tumor progression, metastasis, and angiogenesis were observed. These data have been added to Fig. S4.

On the other hand, as you pointed out, *CASP4/Casp11* is a gene deeply involved in innate immunity. Therefore, it is important to consider its impact on the host tumor immunity. These findings have been reported on page 19, lines 299 – 305 and page 25, lines 401 - 405 of the revised manuscript.

6. Fig 5G, NFKb inhibition also significantly reduced the expression of CASP4 in C4OE cells, suggesting that CASP4 expression is partially driven by activated NFKb signaling. Hence, the decreased expression of PTGS2 and SERPINE1 upon NFKB inhibitor treatment may be a result of reduced expression of CASP4. This need to be clarified.

As rightly pointed out, CASP4 mRNA expression was also significantly suppressed by NFκB inhibitor (BAY11-7082). On the other hand, the expression of CASP4 at the protein level was not influenced by the NFκB inhibitor at this time point. Consequently, the decrease in expression of EPHA2, PTGS2, and SERPINE1 was primarily attributed to the action of the NFκB inhibitor. A Western Blot image has been added to Fig 5H. These findings have been reported on page 20, lines 319 - 322 of the revised manuscript.

7. In Fig7B, the usage of C4OE C258A cells should be more clearly clarified. Are the cell dead upon IFNγ treatment in a form of proptosis? or Ferroptosis? How about the role of Gasdermin D in this process?

We have added the annotation “catalytically inactive” to Fig.7B.

Based on your suggestions, we investigated the form of cell death induced by IFN-γ in detail. IFN-γ increased lipid peroxide, an indicator of ferroptosis, but no difference was observed between mock and C4OE (Fig S7E, F). On the other hand, GSDMD cleavage was more strongly observed in C4OE cells (Fig. 7E). These results suggest that the mode of IFN-γ-induced cell death is mainly pyroptosis. These findings have been reported on page 22, lines 359 – 367 and page 26, lines 423 - 426 of the revised manuscript.

8. Fig 6C shows that low dose of IFNγ promoted the expression of CASP4, is that also the case for high-dose IFNγ treatment (the dosage used in killing assay?). It is a little confused that IFNγ can both promote CASP4 expression and induce cell death in CASP4 overexpressing cells. The author should compare casp4 expression in mock and C4OE cells upon high-dose IFNγ treatment. Moreover, does STAT1 inhibition decrease the sensitivity of C4OE cells to IFNγ induced death?

As pointed out, we think there is some confusion regarding CASP4 induction and cell death induction by IFN-γ. The point we would like to emphasize most is that although IFN-γ induced cell death only at high doses, even low doses strongly induce CASP4 and contribute to cell malignancy. A new diagram has been added to Fig. 6E showing that low doses of IFN-γ induce CASP4 mRNA expression but not cell death, whereas high doses of IFN-γ induce both CASP4 mRNA and cell death.

In addition, we have added a comparison of *CASP4* expression between mock and CASP4 OE under high-dose IFN- γ conditions in Fig. 6F. These findings have been reported on page 21, lines 333 - 339 of the revised manuscript.

We also confirmed that IFN- γ -induced cell death in C4OE was not suppressed under STAT1 inhibitor (fludarabine), but rather the cell death occurred synergistically (Fig 7E). These findings have been reported on page 22, lines 354 – 356 and page 26, lines 426 - 429 of the revised manuscript.

Reviewer #3 (Remarks to the Author):

The title of the manuscript "Caspase-4 promotes both metastasis and interferon- γ -induced cell death in non-small cell lung cancer" promises more than it delivers.

While the paper has novel observations, and the role of caspase-4 in non-small cell subtype adenocarcinoma is well investigated, the presentation of the findings makes it extremely difficult to follow the significance of the manuscript. In its present form, the manuscript can only be appreciated by researchers who work on intracellular signaling in depth. Although the methodology is not particularly novel it is nicely performed.

I believe the manuscript would benefit from rethinking the purpose of the study and reanalysis of some data.

Thank you for your insightful comments. We have revised the manuscript to address the concerns.

1. The manuscript does not investigate the role of caspase-4 in NSCLC in general, but NSCLC-adenocarcinoma.

In the revised manuscript, we have used the expression "lung adenocarcinoma" in place of "NSCLC" in the title.

2. While they collected 77 NSCLC patient samples for the study which I think is a fantastic opportunity to make it clinically relevant, the authors included all the patients. At the same time, some of them were affected by squamous cell carcinoma that was not studied separately.

As rightly pointed out, the 77 NSCLC patients included 56 patients with lung adenocarcinoma, 12 patients with lung squamous cell carcinoma, and 9 patients with other histological types. These data are presented in Figure S1A and Table S1. In addition, we examined the data of non-adenocarcinoma as shown below in response to comment 3.

3. I would expect a figure for patient selection criteria, to focus the study on adenocarcinoma patients.

We narrowed down our analysis to 56 adenocarcinomas from 77 NSCLC patients. Even in adenocarcinoma, *CASP4* mRNA expression was significantly higher in patients with advanced-stage cancer (Stage 3 and 4) (Fig. 1A). On the other hand, no significant difference was observed in

squamous cell carcinoma and other histological types (Fig. S1B). For these reasons, we focused our research on lung adenocarcinoma.

4. I think this would be particularly important, as all the in vitro methodology was performed in lung adenocarcinoma cell lines.

We have added patient selection criteria to Figure S1A and have performed an analysis focused on lung adenocarcinoma patients.

5. I recommend a more detailed, but simple summary figure, to make the results more widely understandable.

Thank you for pointing this out. The summary figure (graphical abstract) has been revised based on the results of the re-experiment. In lung adenocarcinoma, CASP4 contributes to enhanced malignancy and promotes the antitumor effect of IFN- γ via pyroptosis.

6. A more straightforward conclusion is needed

Thank you for your suggestion. Based on your comments, we have revised the conclusion as follows:

Abstract (page 2, lines 14-15) “Our findings indicate that the CASP4 level in primary lung adenocarcinoma can predict metastasis and responsiveness to high-level IFN- γ therapy due to cancer cell pyroptosis.”

Main text (page 27, lines 430-432) “In conclusion, we demonstrated that CASP4 is a tumor angiogenesis and metastasis promoter and a factor that facilitates IFN- γ -mediated pyroptosis. Suppression of CASP4 or IFN- γ therapy are potential therapeutic options for patients with lung adenocarcinoma and high CASP4 expression.”

REVIEWERS' COMMENTS:

Reviewer #1 (Remarks to the Author):

I appreciated the great effort of the authors in the manuscript revision. The paper has been revised according to all my suggestions and comments.

Reviewer #2 (Remarks to the Author):

The authors have satisfactorily addressed all my concerns. In my opinion the revised manuscript has therefore thoroughly dissected the molecular mechanisms underlying the high malignancy of Casp4 high expressing NSCLC cells, the new findings that Casp4 could synergistically promotes proptosis rather than ferroptosis of NSCLC cell would be a bonus, and could add new insights into the potential treatment of lung adenocarcinoma with high Casp4 expression.

Reviewer #3 (Remarks to the Author):

Dear Authors,

I am satisfied with your responses to my questions and comments.

Best wishes,